# Exploring Aleatoric Uncertainty in Object Detection via Vision Foundation Models

## Abstract

Datasets collected from the open world unavoidably suffer from various forms of randomness or noiseness, leading to the ubiquity of aleatoric (data) uncertainty. Quantifying such uncertainty is particularly pivotal for object detection, where images contain multi-scale objects with occlusion, obscureness, and even noisy annotations, in contrast to images with centric and similar-scale objects in classification. This paper suggests modeling and exploiting the uncertainty inherent in object detection data with vision foundation models and develops a data-centric reliable training paradigm. Technically, we propose to estimate the data uncertainty of each object instance based on the feature space of vision foundation models, which are trained on ultra-large-scale datasets and able to exhibit universal data representation. In particular, we assume a mixture-of-Gaussian structure of the object features and devise Mahalanobis distance-based measures to quantify the data uncertainty. Furthermore, we suggest two curial and practical usages of the estimated uncertainty: 1) for defining uncertainty-aware sample filter to abandon noisy and redundant instances to avoid over-fitting, and 2) for defining sample adaptive regularizer to balance easy/hard samples for adaptive training. The estimated aleatoric uncertainty serves as an extra level of annotations of the dataset, so it can be utilized in a plug-and-play manner with any model. Extensive empirical studies verify the effectiveness of the proposed aleatoric uncertainty measure on various advanced detection models and challenging benchmarks.

## 1 Introduction

Deep learning has witnessed remarkable success in a wide range of scenarios and applications for predictive performance, such as image classification Liu et al. (2021); Dosovitskiy et al. (2021); Tolstikhin et al. (2021); He et al. (2016), semantic segmentation Xie et al. (2021); Strudel et al. (2021), and object detection Carion et al. (2020); Zhang et al. (2022); Zhu et al. (2021a); Ren et al. (2015); He et al. (2017). Datasets collected from the open world unavoidably suffer from various randomness or noiseness Kendall & Gal (2017); Cui et al. (2022), resulting in ubiquitous uncertainty inherent in the data (i.e., *aleatoric* uncertainty or data uncertainty Der Kiureghian & Ditlevsen (2009); Hüllermeier & Waegeman (2021)). Quantifying such uncertainty is pivotal for comprehending the inherent fluctuations within the training data, which enables the construction of more resilient models that can accommodate and flexibly respond to conditions characterized by inherent uncertainty.

Compared to images with centric and similar-scale objects in classification benchmarks, images in object detection datasets are typically scene-centric and contain multiple objects in varying scales. Especially, some objects are accompanied by occlusion, obscureness, and even noisy annotations due to limited resources and time in the data collection process Liu et al. (2022) (as observed in Fig. 1). Naturally, the aleatoric uncertainty arises in object detection tasks. However, the majority of aleatoric uncertainty quantification methods target classification or regression problems Kendall & Gal (2017); Chang et al. (2020); Depeweg et al. (2018); Zhang et al. (2024a), with few focusing on the fundamental and challenging object detection. To bridge the gap, we aim to investigate aleatoric uncertainty at the *detection level*, i.e., in the context of object detection.

It is almost impossible for human annotators to compare samples within the dataset and quantify each sample's aleatoric uncertainty due to unaffordable time and resource costs. When discriminative features from object instances are salient and obvious, we consider the aleatoric uncertainty to be low, as such instances can be easily detected and assigned to their semantic classes. However, when

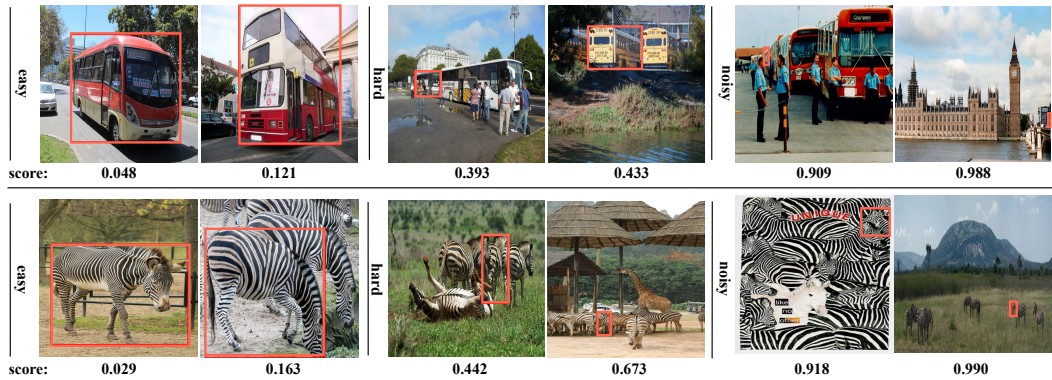

Figure 1: Visualization of scoring objects with corresponding uncertainty scores in training images of MS-COCO. The orange bounding box is the annotated ground truth. "Hard" objects suffer from occlusion or obscureness within an image, and "noisy" ones have misleading bounding boxes.

some of these features are occluded or missing, aleatoric uncertainty increases, making it more challenging to localize and classify such instances. Vision foundation models have learned rich and well-structured features from large-scale training data, enabling them to compare samples from diverse perspectives. In this paper, we opt for SAM Kirillov et al. (2023) as the foundation model and bridge the gap in utilizing SAM to characterize aleatoric uncertainty at the detection level. SAM was trained on the expansive SA-1B dataset Kirillov et al. (2023) that contains more than 1 billion masks spread over 11 million carefully curated images and has established superior performance in addressing open-world vision tasks. Unlike vision foundation models CLIP Radford et al. (2021) and DINOv2 Oquab et al. (2023), SAM received direct supervision in solving dense prediction tasks. Its vision encoder outputs high-resolution feature maps, which is beneficial for processing object detection datasets where objects can vary significantly in size.

In light of this, we proceed to capture the feature of each object instance in the feature space of SAM for measuring aleatoric uncertainty. Building on top of the existing ground-truth bounding boxes and class labels, we perform bounding box-based feature pooling to get a feature vector per object. While SAM was trained in a class-agnostic manner, semantically similar instances are found to be closely crowded together in its feature space. In recent work Xiaoke et al. (2024), one can directly assign semantics and generate captions based on SAM's output embeddings via some text feature mixture and decoder. Based on these observations, we employ a class-conditional Gaussian distribution to model the feature distribution and derive a Mahalanobis distance-based uncertainty score as a measure of aleatoric uncertainty. As shown in Fig. 1, the proposed uncertainty score effectively captures pertinent data characteristics such as object difficulty and noise level, aligning well with human cognitive level.

Furthermore, we devise two practical and crucial usages related to aleatoric uncertainty: uncertainty-aware sample filtering and loss regularization. We can utilize them as proxy tasks to examine the quality of aleatoric uncertainty and enhance detection performance. Firstly, we introduce a quantile function based on aleatoric uncertainty scores to abandon noisy samples that may mislead model training, as well as redundant samples within sub-populations grouped by uncertainty scores to improve training efficiency. Secondly, we propose a sample adaptive training objective that incorporates uncertainty-aware entropy to regularize the binary cross-entropy loss, which can balance easy and hard samples more knowledgeably compared to typical focal loss Lin et al. (2017) and entropy regularization Pereyra et al. (2017).

Aleatoric uncertainty measure can serve as additional annotations of training data thus it can be employed for any model in a plug-and-play way. We conduct extensive empirical studies on challenging benchmarks: MS-COCO Lin et al. (2014) and BDD100K Yu et al. (2020), corresponding to natural and self-driving scenarios, respectively. These studies were performed using various advanced detectors, e.g., CNN-based YOLOX Ge et al. (2021) and FCOS Tian et al. (2019), and transformer-based Deformable DETR Zhu et al. (2021a) and DINO Zhang et al. (2022), to verify the effectiveness of the aleatoric uncertainty measure. We first show that the sample adaptive regularizer incorporated data uncertainty can improve detection performance regarding averaged precision and recall. Furthermore, significant performance gains are observed when aleatoric uncertainty is exploited to abandon noisy samples, and our uncertainty-aware filter strategy outperforms uniform

sampling for redundant instances filtering. Finally, we conduct informative ablation studies to show the robustness of hyperparameters and further explore the potential of aleatoric uncertainty.

## 2 RELATED WORKS

**Wide applications of SAM.** SAM Kirillov et al. (2023) is a vision foundation model designed to address dense prediction tasks by outputting instance masks and parts within regions of interest specified via visual prompts such as points and bounding boxes. Its strong generalization capabilities across domains have enabled a wide spectrum of downstream use cases. While SAM itself only provides class-agnostic masks, it can be utilized after semantic-aware object detection to generate masks for each bounding box. For instance, Grounded-SAM Ren et al. (2024) that connects SAM with Grounding DINO Liu et al. (2023b) is a strong open-world object detection and segmentation model with text prompts. Exploiting caption models Li et al. (2022; 2023a) or image tagging models Huang et al. (2023a); Zhang et al. (2023); Huang et al. (2023b) to get semantic descriptions of images and further use them as text prompts, Grounded-SAM serves as an effective auto-labeling tool. Additionally, SAM has been employed in segmentation tasks within industrial defect segmentation Cao et al. (2023); Li et al. (2024) and medical image segmentation Zhang et al. (2024b).

In recent work Xiaoke et al. (2024), SAM was found to know semantics implicitly. Instead of starting from a semantic-aware object detection model, SAM can do captioning and assign semantic classes to the generated masks through a combination of text feature mixture and a text decoder following its vision encoder. We target a novel use case of SAM: annotating the aleatoric uncertainty of each training sample, which is distinct from the annotations of usual bounding boxes and masks. We benefit from the implicit semantic knowledge in the feature space of SAM's vision encoder. Nevertheless, our use case does not rely on an extra text decoder or feature mixture.

**Feature space density modeling.** Understanding data distribution provides insights into data structure, the generation of additional samples following the same distribution, and out-of-distribution (OOD) detection. Leveraging feature extractors trained to provide compact and informative data representations, feature space density modeling has been proven more effective for tasks like OOD detection, e.g., Kirichenko et al. (2020); Ren et al. (2021); Liang et al. (2022). Based on the familiarity hypothesis in Dietterich & Guyer (2022), relying on rich features is particularly beneficial. Due to their large-scale training sets, the vision encoders of foundation models like SAM effectively fulfill this purpose. While various density modeling techniques have been developed and OOD detection is one of the main use cases, we introduce a new use case: aleatoric uncertainty estimation, which is distinct from OOD detection. Although we employ a standard density modeling method, the achieved gains highlight the potential in this novel application.

**Aleatoric uncertainty.** In deep learning, uncertainty can be classified into two categories: *aleatoric* or data uncertainty and *epistemic* or model uncertainty Der Kiureghian & Ditlevsen (2009); Kendall & Gal (2017); Hüllermeier & Waegeman (2021). Depeweg et al. (2018) propose a decomposition method of uncertainty to capture aleatoric uncertainty from the predictive distribution of Bayesian neural networks with latent input variables. Similarly, Kendall & Gal (2017) developed a technique using MC-dropout Gal & Ghahramani (2016) to independently characterize both uncertainty components. Zhang et al. (2024a) propose a prediction-model-agnostic denoising approach to estimate aleatoric uncertainty for regression by augmenting a variance approximation module under the assumption of the zero mean distribution of data noise. Chang et al. (2020) introduces a data uncertainty-aware method for face recognition by learning feature (mean) and uncertainty (variance) simultaneously in the feature embedding. Prior works mainly estimate aleatoric uncertainty for classification or regression tasks by predictive uncertainty decomposition on task-specific data distribution and training model. This explores the ability of vision foundation models trained on diverse data to be used to quantify data uncertainty from a data distribution perspective.

## 3 ALEATORIC UNCERTAINTY QUANTIFICATION IN OBJECT DETECTION

As data collection and annotation processes inevitably suffer from varying degrees of corruption, aleatoric uncertainty (i.e., data uncertainty) is ubiquitous in real-world datasets. Accurately characterizing data uncertainty can help us better understand training data to utilize it more efficiently and reliably, especially for modern large-scale datasets. To quantify data uncertainty, we leverage SAM to extract the feature of each object instance and model the training data distribution by fitting a multivariate Gaussian distribution in the feature space. Prior work Cui et al. (2024) has shown the effectiveness of Gaussian distribution modeling on classification tasks. We anticipate that easy

samples with low uncertainty will be closely crowded together, while hard/noisy ones with high uncertainty will be far away from the population and more dispersed. A similar intuition is utilized to quantify uncertainty in the classification literature in previous works Van Amersfoort et al. (2020); Mukhoti et al. (2023). From the perspective of density estimation within feature distribution, we derive a Mahalanobis distance-based uncertainty score to represent aleatoric uncertainty. We detail the whole process in Algorithm 1.

**Multivariate Gaussian distribution.**    The training dataset consists of the image-label pairs: $\mathcal{D} = \{(x_i, y_i)\}_{i=1}^N$ with $x_i \in \mathbb{R}^d$ and $y_i = \{b_j, c_j, s_j\}_{j=1}^M$, and $y_i$ represents the set of ground-truths for each image where $b_j \in \mathbb{R}^4$ and $s_j$ is the bounding box and binary mask for each object instance $z_j$, and $c_j \in \{1, \ldots, K\}$ is the corresponding class. Let $V(\cdot)$ denote the feature map layer of the vision encoder in SAM, and we can employ it to obtain each image's feature embedding $V(x_i)$. Building on $V(x_i)$ and corresponding ground-truths: $b_j$ or $s_j$, we further acquire each object's feature vector: $V(z_j)$. The conditional Gaussian distribution with the class $k$ can be defined as:

$$P(V(z) \mid c = k) = \mathcal{N}\left(V(z) \mid \mu_k, \Sigma\right), \tag{1}$$

where $\mu_k$ is the mean vector for class $k$, and $\Sigma$ is an averaged covariance matrix shared by all classes for all training samples. Specifically, we can empirically estimate them by

$$\mu_k = \frac{1}{N_k} \sum_{j:c_j=k} V(z_j),$$

$$\Sigma = \frac{1}{N} \sum_k \sum_{j:c_j=k} (V(z_j) - \mu_k)(V(z_j) - \mu_k)^\top, \tag{2}$$

where $N_k$ is the number of training samples (i.e., object instances) with the label $c_j = k$.

**Mahalanobis distance-based uncertainty score.**    Leveraging the class-conditional Gaussian distributions fitted above, we measure the Mahalanobis distance between training object $z$ and the corresponding class-conditional Gaussian distribution to represent the aleatoric uncertainty of each object in the training set., i.e.,

$$\mathcal{M}(z_j|c_j) = -\left(V(z_j) - \mu_{c_j}\right)^\top \Sigma^{-1} \left(V(z_j) - \mu_{c_j}\right). \tag{3}$$

The Mahalanobis distance $\mathcal{M}(z_j|c_j)$ measures the distance between an object and the centroid of the category $c_j$. A small $\mathcal{M}(z_j|c_j)$ indicates that the object has typical features of the sub-population belonging to this class and boils down to low data uncertainty. Oppositely, the object with the high $\mathcal{M}(z_j|c_j)$ tends to contain ambiguous information (i.e., insufficient identifying characteristic) or noisy annotation (i.e., ambiguous bounding box or even wrong class label).

In order to more conveniently exploit data uncertainty, we employ a scaling procedure to transform the Mahalanobis distance to a range of $(0, 1)$, which is achieved through a combination of log transformation and min-max normalization techniques:

$$d(z_j|c_j) = \frac{\log\left(\mathcal{M}(z_j|c_j)\right) - \min_{j:c_j=k}\{\log \mathcal{M}(z_j|c_j)\}}{\max_{j:c_j=k}\{\log \mathcal{M}(z_j|c_j)\} - \min_{j:c_j=k}\{\log \mathcal{M}(z_j|c_j)\}}, \tag{4}$$

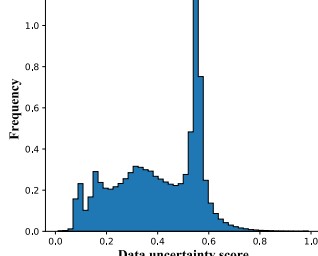

Figure 2: The histogram of $d(z_j|c_j)$ for MS-COCO.

where the Mahalanobis distance belonging to each class is individually normalized to $(0, 1)$. Fig. 2 illustrates the distribution of $d(z_j|c_j)$ for the training data of MS-COCO Lin et al. (2014), and we also show the distribution by categories in Appendix. It is evident that a small percentage (approximately 5%-10%) of samples exhibit high uncertainty scores, implying the presence of noisy objects within the dataset. Additionally, a significant proportion of objects in the MS-COCO dataset are characterized as difficult/hard, as evidenced by the high density of uncertainty scores within the range of $0.5 - 0.6$.

Furthermore, we give some sorting examples and their uncertainty scores belonging to classes "bus" and "zebra" in Fig 1, see Appendix for more visual examples. We can observe a high level of agreement between human visual perception and MD-based data uncertainty scores. In conclusion, our empirical investigation suggests that:

- The low data uncertainty represents an easy sample that can be readily recognized by humans or models due to abundant and unbroken features.

- The objects with medium uncertainty scores are often located in distant positions or partially obscured within an image, posing challenges for accurate classification and detection.

- The objects with high uncertainty often indicate low-quality samples, which may stem from unrecognizable instances or misleading annotated bounding boxes and categories. These instances are prone to being regarded as data noise due to their ambiguity or inconsistency.

---

**Algorithm 1:** Aleatoric uncertainty quantification for object detection

---

**Input:** Training dataset $\mathcal{D}$, the feature map layer of vision encoder in SAM: $V(\cdot)$

1   **for** $x_i, y_i$ *in* $\mathcal{D}$ **do**
2     Get feature embedding $V(x_i)$ for each $x_i$;
3     **for** $b_j, s_j, c_j$ *in ground-truths set* $y_i$ **do**
4       Compute the feature vector $V(z_j)$ of each object based on ground-truths $b_j$ or $s_j$;
5       Add the feature vector $V(z_j)$ to the feature set $\mathbf{V}$;
6     **end**
7 **end**
8 Compute the mean vector and averaged covariance matrix using Eqn. (2);
9 Compute Mahalanobis distance of each object using Eqn. (3);
10 Obtain the final uncertainty score of each object by Eqn. (4) and save all uncertainty scores.

---

## 4   A Recipe for Aleatoric Uncertainty in Object Detection

This section explores the practical usage of aleatoric uncertainty in object detection. Based on estimated aleatoric uncertainty, we propose various data filtering strategies, which aim to remove underlying noisy and redundant objects from the training dataset, leading to more efficient and reliable model training. To further enhance the predictive performance, we develop an uncertainty-aware regularizer and incorporate it into the loss function. Moreover, these two usages can also serve as a proxy for examining the quality of estimated aleatoric uncertainty. It is worth noting that we can calculate the per-object uncertainty score beforehand and treat it as an offline proxy, so the proposed uncertainty-aware usages do not take any additional computational overhead during the model training. In particular, uncertainty scores can serve as an extra level of annotations of the training set and be utilized for any model in a plug-and-play way.

### 4.1   Aleatoric uncertainty-aware data filtering

**Filtering out noisy objects.** As shown in Fig. 1 and 2, some objects have incomplete discriminative features or incorrect annotations, which can damage model training and lead to poor predictive performance. Given this, we propose discarding possible noisy samples that are harmful to model learning. Specifically, we employ a quantile function to discard objects with high uncertainty scores during model training. Let $F$ denote the cumulative distribution function (CDF) of uncertainty scores over all classes, and then we can use the inverse function of CDF $F^{-1} : [0, 1] \rightarrow d(z_j|c_j)$ to represent its quantile function:

$$F^{-1}(p) = \inf \{d : p \leq F(d(z_j|c_j))\} . \tag{5}$$

After that, we retain objects $\mathcal{D}^*$ that are smaller than the specific quantile $p$ (e.g, $p = 95\%$) used for model training, i.e.,

$$\mathcal{D}^* = \left\{ z_j | d_j \leq F^{-1}(p) \right\}_{j=1}^{N*M} . \tag{6}$$

Considering the class-imbalanced issues Lin et al. (2014) in the MS-COCO dataset, we also try discarding noisy objects according to per class, i.e., first calculating the inverse function of CDF of each class $c_j$, referred to as $F_{j:c_j=k}^{-1}(p)$, and then retaining the objects $\mathcal{D}^*$ that meet:

$$\mathcal{D}^* = \left\{ \left\{ z_j | d_j \leq F_{j:c_j=k}^{-1}(p) \right\}_{j:c_j=k}^{N_k} \right\}_{k=1}^{K}, \tag{7}$$

where $N_k$ is the number of object instances with the label $c_j = k$.

**Filtering out redundant objects.** Object detection datasets, such as MS-COCO, typically contain numerous similar objects with common patterns. Thus, an additional useful application of the uncertainty score is eliminating potentially redundant objects from the training set. Objects with closely clustered uncertainty scores within each class often exhibit similar or common patterns. Consequently, the model may only need to learn from a subset of these instances to achieve satisfactory

performance. In this spirit, we can select a certain proportion of objects, known as valuable samples, from each sub-population with close uncertainty scores to enhance training efficiency.

Concretely, we group the uncertainty score of each object into $M$ interval bins for each class (each of size $1/M$) and randomly throw away $p\%$ objects in each bin. We use 10 bins in this work, and we provide the results of more bins in Table A1 in the Appendix. Let $B_m^{c_j}$ be the set of indices of samples with class $c_j$ whose uncertainty score falls into the interval $I_m = \left(\frac{m-1}{M}, \frac{m}{M}\right]$, and the object set that we retain: $\mathcal{D}^*$ can be denoted as:

$$\mathcal{D}^* = \left\{\left\{z_j | j \in B_m^{c_j=k}\right\}_{m=1}^{M}\right\}_{k=1}^{K}, \tag{8}$$

### 4.2 Aleatoric uncertainty-aware Regularization

The uncertainty score serves as a valuable tool for characterizing each object's difficulty and noise level, as demonstrated in Fig. 1. Therefore, it is worth exploring how to leverage this knowledge to enhance model performance. The object detection models usually optimize multiple losses, e.g., $L = L_{\text{cls}} + L_{\text{box}} + L_{\text{obj}}$, and the standard training loss formulation is data uncertainty agnostic. The previous work, such as focal loss Lin et al. (2017), primarily focuses on fitting hard samples and mitigating overfitting to easy samples. It is defined as $L_{\text{FL}} = -(1 - P_t)^{\gamma} \log(P_t)$, where $P_t$ is the model's predictive probability of the ground-truth class and $\gamma$ is a predefined coefficient designed to alleviate the model overfitting to the already confident (i.e., $P_t$ close to 1) majority class. Yet, the focal loss is sensitive to coefficient $\gamma$ and may lead to inappropriate or even harmful regularization for some samples based on the predicted probability.

To address this issue, we incorporate data uncertainty score $d(z_j|c_j)$ into classification loss $L_{\text{cls}}$ and propose an uncertainty-aware entropy to regularize the binary cross-entropy loss. Besides, prior work Mukhoti et al. (2020) has demonstrated that cross-entropy loss equipped with a maximum-entropy regularizer can be interpreted as the lower bound of focal loss, resulting in the ability of the proposed uncertainty-aware entropy regularizer to ensure the optimal performance of the model. As a result, we arrive at the sample adaptive classification loss:

$$\mathcal{L}_{cls} = -\frac{1}{N*M} \sum_{j=1}^{N*M} \left((1 - c_j) \log(1 - f_\theta(z_j)) + c_j \log(f_\theta(z_j)) - \beta d(z_j|c_j)\mathcal{H}[f_\theta(z_j)]\right), \tag{9}$$

where $f_\theta(z_j)$ and $\mathcal{H}[f_\theta(z_j)]$ refer to the predictive binary probability distribution and corresponding entropy for the object $z_j$. $\beta$ is a predefined coefficient to control the strength of entropy regularization, which generally ranges from $(0.1, 0.3)$. Since we can estimate the per-object uncertainty score once before training, the proposed training objective scarcely introduces additional computing overhead.

## 5 Experiments

To verify the effectiveness of the proposed aleatoric uncertainty measure in conveying valuable information about the dataset, we report the predictive performance when employing it for both data filtering and sample adaptive regularization. We mainly present primary experimental results on the challenging benchmark MS-COCO Lin et al. (2014) for the bounding box detection task and use Deformable DETR Zhu et al. (2021b) and anchor-free YOLOX Ge et al. (2021) as the object detection models. In the Appendix, we provide more results for other detection models, such as FCOS Tian et al. (2019) and DINO Zhang et al. (2022). We also validate our method on the challenging self-driving dataset: BDD100K Yu et al. (2020) and other VFMs. Moreover, we ablate the robustness of the proposed uncertainty-aware entropic regularizer to hyper-parameters.

**Datasets.** The 118k train set (train2017) and the 5k validation set (val2017) of COCO 2017 are utilized for training and evaluating the model on the bounding box (bbox) detection task. COCO 2017 comprises 80 classes and encompasses a diverse range of scenes, including indoor and outdoor environments, urban and rural settings, as well as various lighting and weather conditions. The training set contains, on average, 7 instances per image, with a maximum of 63 instances observed in a single image. These instances span a wide range of sizes, from small to large.

**Metrics.** To evaluate the prediction quality, we report averaged precision (AP) and recall (AR) over IoU thresholds, $\text{AP}_{50}$, $\text{AP}_{75}$, and $\text{AP}_L$, $\text{AP}_M$, $\text{AP}_S$ for various-scale objects.

**Implementation Details.** In our study, we mainly employ two typical detection models as detectors: the transformer-based Deformable DETR (trained up to Epoch 50, with a 4-scale setup) and the CNN-based anchor-free YOLOX (specifically, YOLOX-S and YOLOX-M versions), and model details are summarized in Table 1. We also report the performance of more detectors in the Appendix.

Deformable DETR surpasses previous DETR Carion et al. (2020) in both performance and efficiency, achieving better performance than DETR (especially on small objects) with 10×less training epochs by combining the best of the sparse spatial sampling of deformable convolution. YOLOX transforms the traditional YOLO detector, such as YOLOv3 Redmon & Farhadi (2018), into an anchor-free method and enhances it with a decoupled head

Table 1: Model details.

| Model | Params | GFLOPS |
|-------|--------|--------|
| YOLOX-S | 9M | 26.8 |
| YOLOX-M | 25M | 73.8 |
| D-DETR | 40M | 265 |

and the proposed label assignment strategy SimOTA, thereby achieving state-of-the-art performance. As for implementation details, e.g., data preprocessing, experimental settings, etc., we completely follow the original paper. Moreover, we do not use strong data augmentation techniques such as Mixup Zhang et al. (2018) for all experiments. For the hyper-parameters in the proposed training loss 9, we set $\beta$ as 0.2 and 0.3 for YOLOX and Deformable DETR, respectively.

## 5.1 PERFORMANCE ON UNCERTAINTY-AWARE REGULARIZER

Table 2 demonstrates the performance comparison between binary cross-entropy with a constant weighting (Entropy) and uncertainty-aware entropy (UA-entropy) for YOLOX-S, YOLOX-M, and Deformable DETR. The proposed uncertainty-aware entropic regularizer is obviously the top-performing one and leads to a consistent improvement across all detection models. Notably, the performance gain is also prominent for the small-scale models, i.e., YOLOX-S and YOLOX-M, indicating that the proposed data uncertainty measure can convey valuable information about the dataset to model learning. More importantly, the superior performance gain of the proposed sample adaptive regularizer on small-scale models holds significant implications for real-world model deployment. Conversely, regularizing each sample with equal entropy shows only slight improvement or even deteriorates model performance, especially for the small-scale detector YOLOX-S (-0.88% AP). Moreover, the proposed method also achieves significant performance gain on more advanced Deformable DETR with focal loss, implying that the proposed training objective effectively combines data uncertainty to more reasonably balance the learning of difficult and easy samples. We also report the performance of other detectors (i.e., FCOS and DINO) in the Appendix, showing the consistent performance gain.

Table 2: Performance comparison of uncertainty-aware entropy (UA-entropy) and constant entropy regularizer (Entropy) on COCO valset.

| Model | Method | AR | AP | $AP_{50}$ | $AP_{75}$ | $AP_L$ | $AP_M$ | $AP_S$ |
|-------|--------|-----|-----|-----------|-----------|--------|--------|--------|
| YOLOX-S | Vanilla | 53.92 | 39.43 | 57.62 | 42.53 | 52.53 | 43.24 | 21.22 |
| | Entropy | 52.97 | 38.55 | 55.83 | 41.58 | 51.71 | 42.43 | 20.34 |
| | UA-entropy | **54.26** | **39.85** | **58.66** | **43.13** | **52.84** | **43.67** | **22.05** |
| YOLOX-M | Vanilla | 57.92 | 44.34 | 62.27 | 47.98 | 58.32 | 48.33 | 26.69 |
| | Entropy | 58.22 | 44.41 | 62.54 | 48.22 | 58.31 | 48.71 | 26.84 |
| | UA-entropy | **58.86** | **45.33** | **63.78** | **49.12** | **58.99** | **49.94** | **27.97** |
| Deformable DETR | Vanilla | 67.44 | 46.22 | 65.23 | 50.00 | 61.73 | 49.21 | 28.82 |
| | Entropy | 67.23 | 46.10 | 65.01 | 49.25 | 61.06 | 48.34 | 28.17 |
| | UA-entropy | **68.43** | **47.59** | **66.84** | **51.96** | **62.57** | **50.66** | **30.34** |

## 5.2 PERFORMANCE ON UNCERTAINTY-AWARE DATA FILTER

**Filtering of noisy objects.** We verify the effectiveness of measured data uncertainty in filtering out noisy samples from the training set. Table 3 reports the results of discarding samples corresponding to the highest 5% and 10% uncertainty scores (i.e., filtering out possible noisy samples) for different models. We can observe that the predictive performance of each model is improved when samples with high uncertainty scores are abandoned both for 95% data and 90% data settings, which indicates that the reliability of detecting noisy samples in the training data and these samples do not contribute valuable supervision to model training. Therefore, our data uncertainty scores can serve as effective indicators for identifying noisy samples and mitigating the model learning from misleading supervisory information, thereby enhancing predictive performance.

**Filtering of redundant objects.** Afterwards, we examine the effectiveness of the redundant samples filtering by comparing uncertainty-aware and random discarding (i.e., uniformly dropping a certain

Table 3: Performance of filtering out noisy samples using uncertainty scores on COCO 2017 valset. "95%" represents retaining samples less than 95% quantile of aleatoric uncertainty scores.

| Model | Data(%) | AR | AP | $AP_{50}$ | $AP_{75}$ | $AP_L$ | $AP_M$ | $AP_S$ |
|---|---|---|---|---|---|---|---|---|
| YOLOX-S | 100 | 53.92 | 39.43 | 57.62 | 42.53 | 52.53 | 43.24 | 21.22 |
| | 95 | **54.34** | **39.78** | **58.36** | 42.77 | 51.97 | **43.85** | **22.56** |
| | 90 | 53.77 | 39.41 | 58.15 | **42.91** | 51.87 | 43.55 | 21.43 |
| YOLOX-M | 100 | 56.98 | 44.05 | 61.72 | 47.44 | 58.33 | 48.32 | 26.65 |
| | 95 | **58.62** | **44.86** | 63.17 | 48.54 | 58.39 | 49.15 | **27.26** |
| | 90 | 58.44 | 44.84 | **63.22** | **48.61** | **58.72** | **49.51** | 26.54 |
| Deformable DETR | 100 | 67.44 | 46.22 | 65.23 | 50.00 | 61.73 | 49.21 | 28.82 |
| | 95 | **69.52** | **47.31** | **67.14** | **51.24** | 62.55 | **50.53** | 29.73 |
| | 90 | 69.37 | 47.22 | 67.02 | 51.06 | **62.76** | 50.24 | **29.81** |

Table 4: Performance of filtering out redundant samples using uncertainty-aware filter and uniform sampling on COCO 2017 valset. "95%" represents abandoning 5% redundant samples.

| Model | Data(%) | AR | AP | $AP_{50}$ | $AP_{75}$ |
|---|---|---|---|---|---|
| | | | Random / Ours | | |
| YOLOX-S | 95 | 53.92/**54.05** | 37.72/**39.54** | 54.68/**58.37** | 40.66/**43.12** |
| | 90 | 53.85/**53.86** | 36.74/**39.12** | 52.94/**57.88** | 39.83/**42.44** |
| | 80 | 53.51/**53.56** | 36.46/**39.05** | 52.81/**57.94** | 39.59/**42.13** |
| | 70 | 53.15/**53.21** | 35.12/**38.66** | 53.83/**57.53** | 38.18/**41.44** |
| YOLOX-M | 95 | 57.11/**58.68** | 43.22/**45.21** | 60.03/**63.62** | 47.05/**48.95** |
| | 90 | 57.65/**58.03** | 43.04/**44.32** | 60.12/**62.97** | 46.88/**48.21** |
| | 80 | 57.24/**57.44** | 43.01/**44.01** | 60.03/**62.23** | 46.80/**47.32** |
| | 70 | 59.25/**59.54** | 42.52/**43.66** | 60.01/**61.11** | 46.44/**46.65** |
| Deformable DETR | 95 | 68.03/**68.36** | 46.01/**46.40** | 65.12/**65.76** | 49.85/**50.33** |
| | 90 | 67.51/**68.27** | 45.59/**46.05** | 64.06/**65.99** | 49.13/**49.94** |
| | 80 | 66.24/**67.49** | 44.26/**45.47** | 63.08/**65.44** | 48.13/**49.07** |
| | 70 | 65.47/**66.71** | 43.75/**44.82** | 62.30/**64.25** | 47.27/**48.20** |

percentage of samples) strategies, with the experimental results summarized in Table 4. As shown, the proposed uncertainty-aware filtering strategy consistently outperforms uniform sampling for all metrics under different data percentages, suggesting that leveraging data uncertainty scores to cluster samples (i.e., grouping overall training data into multiple subsets with similar patterns) is reliable. Furthermore, uniforming data selection can dramatically degrade predictive performance on relatively small-capacity models like YOLOX-S. Oppositely, our uncertainty-aware data sampling still maintains superior performance, with only a marginal reduction of 0.8% in AP while discarding 30% of the data. Interestingly, uncertainty-aware data sampling with 95% data surpasses predictive performance with 100% data, which further verifies the existence of noisy samples in training data. In the future, the proposed uncertainty-aware filtering could serve as a new paradigm for data pruning.

### 5.3 ABLATION STUDIES

This section further examines the effectiveness of our aleatoric uncertainty measure on the self-driving dataset. We also conduct ablation studies on different vision backbones (e.g., DINOv2), the hyper-parameters $\beta$ in Eqn. 9 and the combination of aleatoric uncertainty-aware filter and regularizer.

Table 5: Results (AP) on the self-driving dataset: BDD100K. "95% data" denotes abandoning 5% samples with the highest uncertainty scores, and the same meaning goes for "90% data".

| | Vanilla | UA-entropy | 95% data | 90% data |
|---|---|---|---|---|
| YOLOX-S | 28.16 | **32.53** | **33.41** | **33.38** |
| YOLOX-M | 30.17 | **34.02** | **34.15** | **34.20** |
| D-DETR | 65.33 | **68.72** | **69.02** | **68.81** |

**Effectiveness on the self-driving dataset.** We further examine the performance of the proposed data uncertainty measure for object detection in the self-driving scenario using the BDD100K dataset Yu et al. (2020). This large-scale and long-tailed driving video dataset includes a diverse

set of 100k annotated images (70k/10k/20k images for train/val/test set) with 10 classes for object detection. Table 5 presents the performance of data uncertainty scores used for entropy regularization and noisy sample filtering, showing significant gains in terms of average precision (AP) on YOLOX and Deformable DETR. We can especially observe a more prominent gain on small-scale YOLOX-S. All of this further confirms the superior scalability of our method across different real-world tasks.

**Different vision backbones.** We further examine the applicability of our approach to additional vision foundation models, such as SAM2 and DINOv2 Oquab et al. (2023). Due to the inherent resolution limitation of DINOv2, we incorporate LoftUp Huang et al. (2025) (Learnable Feature Upsampling)—a recent technique designed to enhance the spatial resolution of features extracted from vision backbones—to upsample its feature maps before applying our framework. Table 6 reports the performance of UA-entropy with DINOv2 enhanced by LoftUp as well as SAM2 on the COCO 2017 val set. As shown, our method consistently improves performance across both backbones, further demonstrating its generalizability and effectiveness.

Table 6: Performance of UA-entropy with DINOv2 and SAM2 on COCO 2017 valset.

| Model | Method | DINOv2 | | SAM2 | |
|---|---|---|---|---|---|
| | | AR | AP | AR | AP |
| YOLOX-S | Vanilla | 53.91 | 39.43 | 53.91 | 39.43 |
| | UA-Entropy | **54.31** | **39.83** | **54.49** | **39.85** |
| YOLOX-M | Vanilla | 57.92 | 44.34 | 57.92 | 44.34 |
| | UA-Entropy | **58.90** | **45.39** | **58.96** | **45.45** |

**Regularization coefficient $\beta$.** We analyze the effect of hyper-parameter $\beta$ on predictive performance in the loss function 9. Table A2 reports the comparison results under various $\beta$ for constant weighting and uncertainty-aware entropy regularization on YOLOX-S. As shown, the entropy penalty with a constant weighting is particularly sensitive to hyper-parameter $\beta$, with large values (e.g., 0.4) resulting in significantly poor performance. In contrast, the proposed data uncertainty-aware regularizer is robust to $\beta$ owing to sample-uncertainty adaptive weighting, which highlights that data uncertainty provides a more reliable way to balance difficult and easy samples.

**Combination of uncertainty-aware data filter and regularization.** Table 2 and 4 have shown the effectiveness of data uncertainty for redundant sample filtering and entropy regularization. It is worth exploring whether the performance of uncertainty-aware data filtering can be further enhanced by incorporating sample-adaptive regularization. To this end, we compare the predictive performance of using uncertainty-aware data filtering alone versus its combination with sample-adaptive regularization under different proportions of training data. As shown in Fig. A4 in Appendix, the proposed sample-adaptive regularization (UA-entropy) consistently improves the performance of redundant sample filtering on YOLOX-S and YOLOX-M by incorporating data uncertainty of each object to adaptively balance the impact of easy and hard samples within the remaining data.

## 6 CONCLUSIONS

This work investigates an important yet under-explored problem – how to accurately characterize aleatoric uncertainty in object detection. Profiting from the powerful feature representation capabilities of vision foundation models, we propose to estimate the aleatoric uncertainty of each object based on the representation space of foundation models. Furthermore, we explore two practical uncertainty-related tasks: aleatoric uncertainty-aware sample filtering and loss regularization. These tasks serve a dual purpose: examining the quality of aleatoric uncertainty and being used to develop a data-centric learning paradigm aimed at enhancing model performance and training efficiency. Extensive empirical studies validate the effectiveness of the proposed aleatoric uncertainty measure, demonstrating consistent performance gains across various advanced detection models.

In the future, we can explore leveraging various vision foundation models, e.g., DINOv2 Oquab et al. (2023) and GroundingDINO Liu et al. (2023b), to quantify data uncertainty at the detection level. Additionally, it is critical to develop more techniques, such as knowledge distillation, to extract valuable knowledge from foundation models for uncertainty quantification. Large Vision Language Models (LVLMs) Liu et al. (2024); Zhu et al. (2023) bridge the gap between visual and linguistic understanding and exhibit the potential towards achieving general artificial intelligence. However, they also easily produce hallucinations or generate inconsistent responses with input images Liu et al. (2023a); Zhou et al. (2023); Li et al. (2023b). Typically, LVLMs are fine-tuned on language-image instruction-following data generated from COCO, so the proposed noisy sample filtering strategy could be beneficial in enhancing robustness and mitigating hallucinations.

**Reproducibility Statement**   We have made significant efforts to ensure the reproducibility of our work. The details of model architectures, training settings, and evaluation protocols are provided in the main paper. Additional implementation details, hyperparameter configurations, and ablation results are included in the appendix. Furthermore, the complete source code and scripts necessary to reproduce our experiments are provided in the supplementary materials.

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

Table A1: AP of filtering out 10% redundant samples with different bins on COCO 2017 valset.

| Models / Bins | 6 | 8 | 10 | 12 | 16 |
|---|---|---|---|---|---|
| YOLOX-S | 39.04 | 39.10 | **39.12** | **39.14** | 39.07 |
| YOLOX-M | 44.19 | 44.29 | **44.32** | 44.31 | 44.21 |
| Deformable DETR | 45.91 | 46.03 | **46.05** | 46.02 | 45.96 |

## A  HISTOGRAMS OF ALEATORIC UNCERTAINTY SCORES BY CATEGORIES

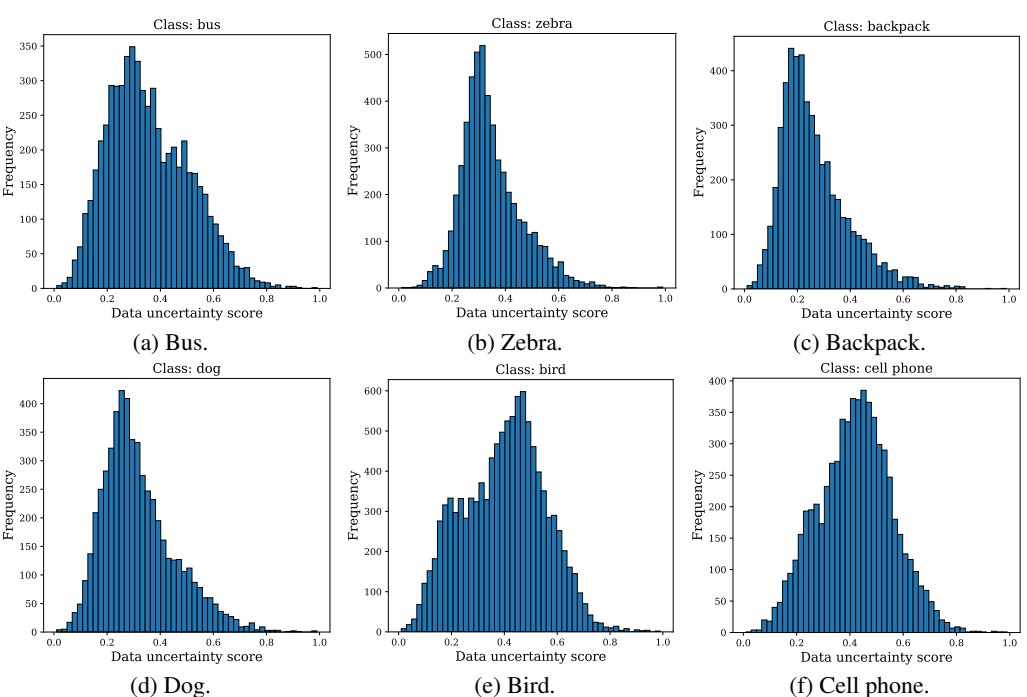

(a) Bus.   (b) Zebra.   (c) Backpack.

(d) Dog.   (e) Bird.   (f) Cell phone.

Figure A1: Distributions of data uncertainty scores for different classes.

## B  MORE SORTING EXAMPLES VIA ALEATORIC UNCERTAINTY ON MS-COCO

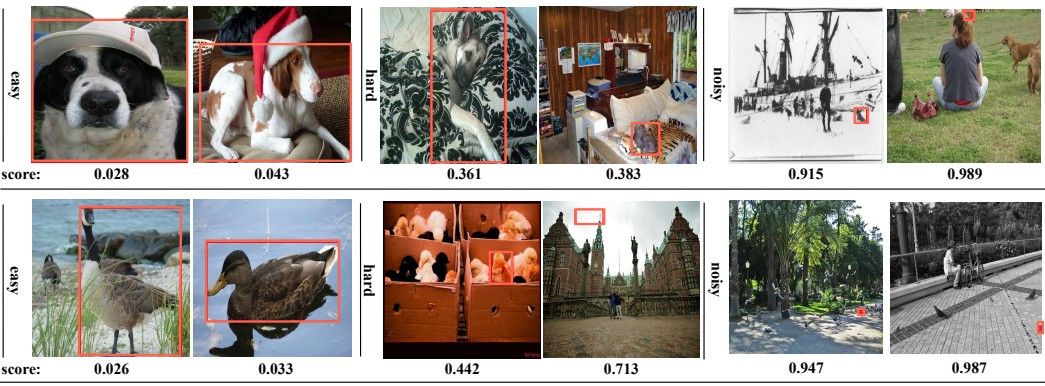

Figure A2: Visualization of scoring objects with corresponding uncertainty scores in training images of MS-COCO Lin et al. (2014) for class "dog" and "bird". The orange bounding box is the annotated ground truth. "Hard" objects suffer from occlusion or obscureness within an image, and "noisy" ones have misleading bounding boxes.

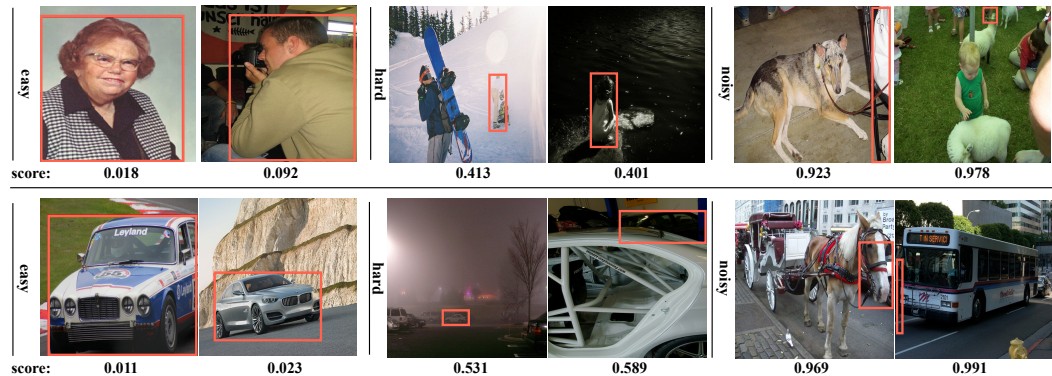

Figure A3: Visualization of scoring objects with corresponding uncertainty scores from training images of Pascal VOC Everingham et al. (2015) for class "person" and "car".

Table A2: The comparison for AP and AR under different $\beta$ on YOLOX-S. **Bold** indicates the results from the chosen hyperparameter.

| $\beta$ | | 0 | 0.10 | 0.20 | 0.25 | 0.30 | 0.40 | 0.50 |
|---|---|---|---|---|---|---|---|---|
| Entropy | AP | 39.43 | 37.22 | **38.55** | 37.75 | 37.47 | 36.85 | 36.03 |
| | AR | 53.92 | 51.88 | **52.97** | 52.10 | 51.90 | 51.02 | 50.42 |
| UA-entropy | AP | 39.43 | 39.75 | **39.85** | 39.81 | 39.77 | 39.54 | 39.44 |
| | AR | 53.92 | 54.17 | **54.26** | 54.24 | 54.02 | 53.88 | 53.91 |

Table A3: Performance comparison of uncertainty-aware regularizer (UA-entropy) and constant entropy regularizer (Entropy) on COCO 2017 valset.

| Model | Method | AR | AP | $AP_{50}$ | $AP_{75}$ | $AP_L$ | $AP_M$ | $AP_S$ |
|---|---|---|---|---|---|---|---|---|
| | Vanilla | 57.21 | 41.46 | 60.71 | 45.08 | 51.53 | 44.82 | 24.41 |
| FCOS | Entropy | 57.10 | 41.35 | 60.62 | 45.01 | 51.61 | 44.24 | 24.03 |
| | UA-entropy | **58.62** | **42.62** | **62.13** | **46.36** | **52.74** | **45.67** | **25.33** |
| | Vanilla | 72.63 | 49.39 | 66.97 | 53.84 | 63.64 | 52.30 | 32.48 |
| DINO | Entropy | 72.23 | 49.30 | 66.78 | 53.11 | 63.06 | 51.65 | 32.15 |
| | UA-entropy | **73.59** | **49.59** | **67.01** | **54.23** | **63.67** | **53.07** | **32.54** |

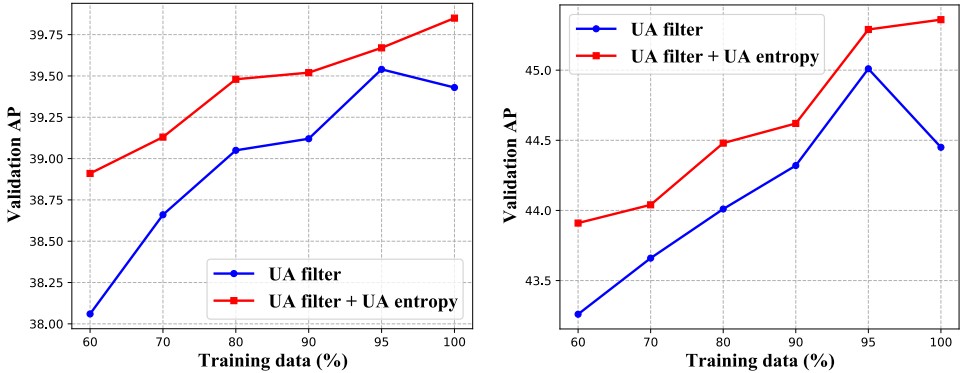

(a) The performance comparison on YOLOX-S.   (b) The performance comparison on YOLOX-M.

Figure A4: The effect of combining uncertainty-aware redundant samples filtering (UA-filter) and regularization (UA-entropy) on performance on YOLOX-S and YOLOX-M.

