# OpenReview forum: "Exploring Aleatoric Uncertainty in Object Detection via Vision Foundation Models"
_ICLR.cc/2026/Conference — ICLR 2026 Conference Withdrawn Submission_

### Official Review · Reviewer_2soU · 2025-10-28

**Soundness:** 1
**Presentation:** 2
**Contribution:** 1
**Rating:** 2
**Confidence:** 3

**Summary:**

This paper addresses the aleatoric uncertainty that naturally arises in open-world datasets, which often contain randomness and noise. Focusing on object detection which involves occlusion, scale variation, and noisy labels, the authors propose a data-centric reliable training paradigm using vision foundation models (VFMs). The authors model object features from VFMs with a mixture-of-Gaussians and computing Mahalanobis distance-based measures, showing the possible practical usages: uncertainty-aware sample filtering and sample-adaptive regularization.

**Strengths:**

- The authors test their algorithm in various scenarios using two base models, two datasets, and three VFMs to demonstrate its robustness.
- The authors clearly shows the effectiveness of uncertainty-aware use cases in the evaluation section.

**Weaknesses:**

1. Lack of novelty in motivation and solution: Aleatoric uncertainty in object detection has already been explored for quite some time, and for essentially the same reasons. Since what to solve has already been addressed in various ways, this paper instead emphasizes how to solve the problem (by leveraging vision foundation models) which also does not sound particularly novel.

2. Overly verbose and bottom-up writing style (below are two examples):
	•	Lines 73–80: SAM is not the main contribution. Including details such as “11 million” or “1 billion” in the Introduction is unnecessary, as this section should focus on introducing the main storyline. You could simply mention that you aim to leverage SAM’s strong understanding capability through its high-resolution feature maps.
	•	Lines 80–90: The high-level solution description is missing. What exactly enables you to leverage SAM’s capability? What motivates the observation in Line 84? How do you derive the proposed distance metrics? There is no clear high-level explanation of how you conceptualize the problem or what your core intuition or hypothesis is.


3. The authors assume a mixture-of-Gaussians structure for the object features. Is this assumption valid in object detection scenarios (unlike image classification, which involves more centric and similar-scale objects)?

4. Citation format: Use ~\citep or ~\citet instead of ~\cite to improve readability.

5. I am not convinced by the author's design choices: Using large VFMs within Deformable DETR and YOLO architectures seems computationally expensive (VFM size >> DETR/YOLO size). The authors essentially borrow the capabilities of VFMs in the training logic without addressing the fundamental limitations of object detection models. It is unclear whether this approach is truly practical or beneficial.

**Questions:**

1. Section 4.1: Regarding the filtering of noisy and redundant objects: could you provide failure cases (e.g., false positives and false negatives in filtering)? If possible, please also include qualitative examples of both successful and failed cases.
2. Computation Overhead: Please discuss the overhead introduced by per-object uncertainty score computation.
3. Did you observe any notable differences when applying this uncertainty-aware algorithm to both anchor-based object detection models (e.g., Deformable DETR) and anchor-free models (e.g., YOLO)?

---

### Official Review · Reviewer_SzDy · 2025-10-29

**Soundness:** 1
**Presentation:** 3
**Contribution:** 1
**Rating:** 4
**Confidence:** 3

**Summary:**

This paper focuses on exploring aleatoric uncertainty in object detection using vision foundation models. It proposes a method that leverages SAM for feature extraction, models uncertainty with a mixture of Gaussian distributions and Mahalanobis distance, and applies the uncertainty scores for sample filtering and loss regularization. Evaluated on datasets like MS-COCO and BDD100K with detectors such as YOLOX and Deformable DETR, it demonstrates some effectiveness but lacks significant innovation, as the core uncertainty quantification and application strategies are mostly adaptations of existing techniques. The performance improvements are marginal, and the experimental comparisons and theoretical analyses are insufficient.

**Strengths:**

1. This paper addresses a practical need in object detection by focusing on aleatoric uncertainty in complex scenarios and follows a clear and reproducible technical path using mature models like SAM and common detectors.
2. This paper conducts relatively comprehensive experiments across datasets, detector architectures, and backbones.

**Weaknesses:**

1. It lacks significant innovation as core methods are adaptations of existing techniques, which can be found in “Mahalanobis Distance for OOD Detection (Lee et al., NIPS 2018)”.

2. It has marginal performance improvements, and insufficient experimental comparisons and theoretical analyses. Specifically, although the authors experimented extensively, the key performance indicator (AP) shows only marginal gains, typically about 0.5% AP over the baseline. This small improvement does not provide sufficient evidence to justify the value and necessity of introducing a new, purportedly innovative method.

**Questions:**

1. Why does it lack comparisons with existing object detection uncertainty quantification methods? In my opinion, the paper claims to address Aleatoric Uncertainty, yet fails to compare its approach against commonly used uncertainty methods applicable to object detection, such as the classic Monte Carlo Dropout (MC-Dropout) or Deep Ensembles. This comparison is necessary to demonstrate the unique advantages of the proposed method in quantifying uncertainty for object detection.

2. The paper fails to provide in-depth analysis or ablation studies concerning object scale variance, which is a critical challenge in object detection. The authors must quantify and report whether their uncertainty scores accurately reflect the intrinsic uncertainty of small and ambiguous objects. The absence of this scale-based robustness analysis significantly undermines the method's practical persuasiveness in real-world detection scenarios.

---

### Official Review · Reviewer_ahvk · 2025-10-31

**Soundness:** 2
**Presentation:** 3
**Contribution:** 2
**Rating:** 4
**Confidence:** 3

**Summary:**

This paper proposes using vision foundation models (specifically SAM) to estimate aleatoric uncertainty in object detection datasets. The authors fit class-conditional Gaussian distributions in SAM's feature space and compute Mahalanobis distance-based uncertainty scores. These scores are then used for: (1) filtering noisy/redundant training samples, and (2) uncertainty-aware entropy regularisation during training. Experiments on MS-COCO and BDD100K show modest improvements across several detectors.

**Strengths:**

1. This is an important problem. Characterising data uncertainty in object detection is a valuable and under-explored problem, particularly given the prevalence of noisy annotations and occluded objects.
2. The approach is practical. The plug-and-play nature of the method is appealing - uncertainty scores can be computed offline and used with any detector.
3. Experimental evaluation is comprehensive. The paper includes experiments across multiple detectors (YOLOX, Deformable DETR, FCOS, DINO) and datasets, showing consistent improvements.
4. Figure 1 provides compelling visual evidence that the uncertainty scores align with human intuition about sample difficulty.

**Weaknesses:**

The core technical contribution is applying existing Mahalanobis distance-based uncertainty estimation (Van Amersfoort et al. 2020, Mukhoti et al. 2023) to SAM features for object detection. This is primarily an application rather than a methodological contribution. The technique of modelling feature distributions with Gaussians and computing Mahalanobis distances is well-established in OOD detection and uncertainty quantification literature.

Several aspects of the theoretical justification are weak. The paper does not convincingly argue why Mahalanobis distance in SAM's feature space specifically measures aleatoric rather than epistemic uncertainty. Many "hard" examples (e.g. occluded objects) could be considered epistemic from a model's perspective. SAM was trained for class-agnostic segmentation, not uncertainty estimation. Why should its feature space be the right representation for quantifying data uncertainty in object detection? The claim of "implicit semantic knowledge" needs stronger empirical validation. Using a shared covariance matrix across all classes (Eq. 2) is a strong assumption that is not well justified. Different object classes likely have different feature variances and correlations.

The paper treats "hard samples" and "uncertain samples" as equivalent, but these are distinct concepts. A hard but correctly labeled occluded object is challenging but not necessarily uncertain. Filtering such samples (Table 3-4) may remove valuable training data. The three-way categorisation into "easy/hard/noisy" is subjective and not rigorously defined.

Improvements are modest and in some cases unvalidated. Performance gains are small (e.g., +0.42% AP for YOLOX-S in Table 2) and no statistical significance testing is provided. The "noisy sample filtering" experiments (Table 3) don't verify that filtered samples are actually noisy. The improvements could simply result from removing hard samples that the model overfits to. Manual verification of filtered samples is needed.

Baselines are weak and some comparisons are missing. The "constant entropy" baseline (Table 2) is weak. More thorough comparison with focal loss variants is needed. There is no comparison with other uncertainty quantification methods for object detection or learning-based uncertainty estimation approaches and there is limited comparison with other vision foundation models (only DINOv2 briefly in ablation).

There are also some more minor issues:
- The process of extracting features from bounding boxes in SAM's feature maps lacks detail. How are features aggregated for objects at different scales?
- While mentioned as "negligible," the computational cost of extracting features for all training objects is not quantified.
- The log transformation and normalisation in Eq. 4 appear arbitrary. Why logarithm specifically? The quantile threshold p and regularisation coefficient beta need more thorough sensitivity analysis.
- Section 4.1's claim that samples with similar uncertainty are "redundant" lacks justification. Similar uncertainty doesn't imply similar features or redundancy.
- Which SAM encoder layer is used? Does this choice matter?
- There could be an ablation on class-specific vs. shared covariance matrices

**Questions:**

Besides responding to the above listed weaknesses the following additional questions could be responded to:

1. Can you provide evidence that filtered samples in Table 3 are actually mislabeled rather than just hard?
2. How does uncertainty score correlate with actual detection errors on validation data?
3. Why is shared covariance better than class-specific covariances?
4. Have you compared with other pre-trained feature spaces (e.g., supervised ImageNet features)?
5. What is the computational overhead of feature extraction for large-scale datasets?

---

### Note · Authors · 2025-11-27

I have read and agree with the venue's withdrawal policy on behalf of myself and my co-authors.